



# Air-sea interaction heat and momentum fluxes based on vessel's experimental observations over Spanish waters

Ángel Sánchez-Lorente [1], Elena Tel [1], Lucía Sanz-Pinilla [1], and Gonzalo González-Nuevo González [2]

[1] Instituto Español de Oceanografía, Servicios Centrales, C/Corazón de María, 8, 28002 Madrid, Spain
[2] Instituto Español de Oceanografía, Centro Oceanográfico de La Coruña, P. Marítimo Alcalde Francisco Vázquez, 10, 15001 La Coruña, Spain

**Correspondence:** Ángel Sánchez-Lorente (angel.sanchez@ieo.csic.es)

**Abstract.** Ocean and atmosphere are directly communicated through air-sea interaction fluxes. These include heat exchange, by latent (*LHFL*) and sensible (*SHFL*) heat, and momentum (*MOFL*), by wind stress. They stand as the leading predictors of how ocean influences atmospheric variability, and vice versa. In this paper, meteorological and upper-ocean measurements collected during the period 2011-2023 aboard four research vessels over Spanish waters and adjacent seas, are used to study

air-sea interaction fluxes. These research vessels are: *Ramón Margalef* (RM), *Ángeles Alvariño* (AJ) and *Cornide de Saavedra* (CS) belonging to *Instituto Español de Oceanografía* (IEO-CSIC); and *Miguel Oliver* (MO), from *Secretaría General de Pesca* of Spain. Recorded data are daily sent to the IEO Data Center on land, where quality control procedures are applied. Heat and momentum fluxes products are derived using the bulk aerodynamic approximation and are stored within a MEDAR/MedAtlas format alongside the meteorological and ocean variables used in the calculation. The data set generated and described here

are publicly available at SEANOE. In order to study the behaviour of these air-sea fluxes, several marine regions subdivisions, based on the heterogeneity of the Spanish waters, are proposed. Additionally, some results within these regions context are shown.

## 1   Introduction

Ocean and atmosphere maintain a close and constant relationship through numerous process that are responsible for countless
effects at different scales, from general circulation of both media to the formation of convective air masses by turbulent process in the atmospheric boundary layer (ABL) (Large and Pond, 1981). The study of these phenomena must focus on air-sea interactions occurring in the interface of both subsystems, which are commonly known as heat and momentum turbulent fluxes. The former refers to sensible and latent fluxes, which account for the loss of heat by the sea due to convection and evaporation, respectively; while the latter is governed by the wind stress at the sea surface. A deeper understanding of the temporal
behaviour, spatial distribution or climatological tendencies will enable a further improvement in understanding climate system and its variability (Ruiz et al., 2008).

The experimental record of these magnitudes is generally scarce. Direct measurement requires very costly and technologically demanding equipment, which can be substituted by several approximations and parameterizations that have been



developed and studied on offshores platforms and vessels in open ocean since the latest decades of the 20th century (Smith, 1980; Large and Pond, 1981, 1982; Grachev and Fairall, 1997; Edson et al., 2013). In most cases, the main purpose of these studies were to calibrate and test the different methods for estimating turbulent fluxes at the air-sea interface under various weather and sea conditions. The experimental record, nevertheless, implies several drawbacks such as data-sampling density, short data series, incomplete spatial coverage or systematical measurement errors.


Recently, other alternative procedures have been developed to better capture air-sea interactions. In this context, satellite-based flux products (Yu and Weller, 2007) have enabled improvements in temporal and spatial resolutions as well as enhanced global coverage; for instance, Kubota et al. (2002) and Bentamy et al. (2003) for sensible and latent fluxes and O'Neill (2012) for wind stress. However, these products are not independent of field measurements due to the need of validation and other
calibration procedures. Additionally, they present certain technical difficulties making it less reliable to derive variables like specific-humidity and air temperature, which ultimately affects heat fluxes outcomes. On the contrary, wind stress is well represented thanks to the finer temporal and spatial resolution and the reduced uncertainties related to record errors (Pierson Jr, 1990). Furthermore, new modelling-based products from various climate-research organisations around the world have contributed to the study of global air-sea variability and its implications, reducing the constraints of field measurements by com-
bining through statistical analysis processes the information from numerous monitoring sources (e.g, satellite, buoys, ships, reanalysis). Notable air-sea interaction products with global coverage include the NCEP Climate Forecast System Reanalysis (CFSR), Objectively Analyzed Air-Sea Fluxes (OAFlux) (Yu and Weller, 2007), the Modern-Era Retrospective Analysis for Research and Applications (MERRA) (Bosilovich, 2008), and the SeaFlux data product (Clayson et al., 2014). However, due to the different analysis processes, and the use of independent data sources, they exhibit discrepancies and biases, unveiling
the need for further improvement of the validation of bulk variables (Bentamy et al., 2017). As an example, turbulent fluxes behaviours appear to be more homogeneous in tropical latitudes compared to mid-latitude and subpolar regions across several analysis products. Additionally, OAFlux and NCEP show great biases in complex circulation areas (Mao et al., 2021), as well as seasonal biases (Zhou et al., 2018), which must be taken into account prior to any subsequent analysis.

Concerning waters in the Mediterranean and Cantabrian-Atlantic areas, there are also previous researches based on dynamical downscaling of the NCEP/NCAR global reanalysis (HIPOCAS) (Vargas-Yáñez et al., 2007; Ruiz et al., 2008). It achieves a spatial resolution of 0.5°x0.5° in the Mediterranean Sea during a 44-year period between 1958-2001. It follows the line of other authors focused on this basin (Bunker et al., 1982; Garrett et al., 1993; Castellari et al., 1998), improving the representation of regional aspects of heat fluxes thanks to its reanalysis methodology. Nevertheless, different resolutions and model configura-
tions introduce biases and discrepancies that require observational data for their validation.

In this line, this paper presents the air-sea interaction fluxes obtained through meteorological and from thermosalinograph (TSG) experimental records during the period 2011-2023 aboard four research vessels: *Ramon Margalef* (RM), *Ángeles Alvariño* (AJ), *Miguel Oliver* (MO) and *Cornide de Saavedra* (CS). These air-sea interaction fluxes are sensible (*SHFL*) and



latent (*LHFL*) heat fluxes, and momentum flux (*MOFL*), also known as wind stress. AJ and RM are part of the current fleet of the *Instituto Español de Oceanografía* (IEO-CSIC), whereas CS is a former vessel currently decommissioned. MO belongs to *Secretaría General de Pesca* of Spain. These multidisciplinary research vessels are equipped with state-of-the-art technological equipment for navigation and for scientific research. Despite the specific objectives of each cruise, continuous meteorological and TSG data are collected during their trajectories and activities. Generally, monitoring sensors are switched off when arriv-

ing at port for preservation and maintenance. However, the high sample frequency during their activities allows to establish continuous temporal series throughout the whole vessel's trajectories.

The paper is organised as follows: firstly, in Sect. 2 a description of the marine areas covered by the vessel's trajectories is shown. Next, in Sect. 3, the description of the data records, the validation and quality control (QC) procedures for the different

data sources, and the methodology employed in the calculation of air-sea fluxes are explained. Some results and discussions are presented in Sect. 4. Additionally, the availability and access of the data is detailed in Sect 5. Lately, a conclusion is provided in the last section.

## 2  Study area

During their activities and trajectories, the AJ, RM, MO and CS vessels cover a large part of the Spanish waters and adjacent

seas. These are notoriously heterogeneous waters, from subtropical to mid-latitude zones. They present different ranges of temperature and salinity due to the constrictions of continental distribution around the specific areas -e.g the semi-enclosed regime of the warm and salty Mediterranean sea in comparison with the open and colder waters of the Cantabrian sea at higher latitudes; or with the subtropical waters of the Canary islands region (Talley, 2011)-. Also, different regional wind regimes, so important for the air-sea interaction, are characteristic for each region (Viedma Muñoz et al., 2005; Azorin-Molina et al., 2018;

Ortega et al., 2023). Thus, the ocean-atmosphere interaction processes present disparities depending on the oceanographic and meteorological characteristics of each location. Therefore, prior to any scientific analysis, such heterogeneity encourages a solid and well-founded subdivision in pertinent marine regions. In Fig. 1 several subdivisions around the Spanish marine territory are proposed. The first five, North Atlantic, Canary, South Atlantic, Alboran-Strait and Levantine-Balearic, are part of the Spanish jurisdictional waters. They are also known as marine demarcations, defined by the impulse of the Marine Strategy

Framework Directive from the EU (Long, 2011) for protection and preservation purposes to prevent marine ecosystems from their deterioration. Additionally, three more regions are offered: Western Mediterranean, which covers the Alboran-Strait and Levantine-Balearic areas; Cantabrian-Atlantic, which involves North Atlantic region alongside northern waters of Vizcaya Gulf; and Atlantic, which extends from the coast of Portugal to the margins of Africa and the Canary islands region.



| Region | Lon W (°) | Lon E (°) | Lat N (°) | Lat S (°) |
|---|---|---|---|---|
| North Atlantic | -13.83 | -1.78 | 46.87 | 41.38 |
| Canary | -21.90 | -11.81 | 32.25 | 24.55 |
| South Atlantic | -7.54 | -5.93 | 37.21 | 37.17 |
| Alboran-Strait | -5.93 | -1.69 | 36.85 | 35.73 |
| Levantine-Balearic | -2.19 | 6.30 | 42.75 | 36.20 |
| Western Mediterranean | -6.00 | 6.50 | 43.00 | 35.70 |
| Cantabrian-Atlantic | -13.83 | -1.16 | 47.8 | 41.38 |
| Atlantic | -13.83 | -7.54 | 41.39 | 32.24 |

**Figure 1.** Marine subdivisions of the different areas covered by AJ, RM, MO and CS vessels' trajectories. Geographical limits specified. Top five regions belong to the Spanish jurisdictional waters within the Marine Strategy Framework Directive from the EU (Long, 2011).

## 3 Methodology

### 3.1 Data records

Four vessel's data records are used to obtain the air-sea interaction fluxes. AJ, RM, MO and CS research vessels are provided with meteorological and sea surface monitoring equipment, allowing to measure the variables required to obtain the final fluxes products throughout their trajectories. The on-board measuring instrumentation is a Vaisala AWS430 unit, specifically designed to measure climatological variables within a marine environment. Information about the instruments used for the measurement of each magnitude is detailed in Table 1. The variables recorded during the period 2011-2023 and stored within this data set are: air temperature (DRYT), atmospheric pressure (ATMS), wind speed (WSPD), relative humidity (RELH) and total incident radiation (RDIN) in the meteorological equipment; and sea temperature (TEMP) using the TSG.

**Table 1.** Vaisala AWS430 meteorological and TSG instrumentation aboard for the experimental record.

| Data Type | Magnitude | Instrument | Description |
|---|---|---|---|
| | DRYT (° C) | HMP155 HUMICAP | Humidity and temperature probe. Heated sensor to avoid condensation and mantain humidity levels within the ambience levels. |
| | ATMS (hPa) | PTB330 | Barometer. A capacitive absolute pressure sensor made of silicon. |
| Meteorological | WSPD ($ms^{-1}$) | WMT700 | Ultrasonic anemometer. Measures the time it takes for the ultrasound to travel from one transducer to another. |
| | RELH (%) | HMP155 HUMICAP | Humidity and temperature probe. |
| | RDIN ($Wm^{-2}$) | CMP3 | Solar radiation pyranometer. Long waves range between 300 and 2800 nm. |
| TSG | TEMP (° C) | DTS12W | Temperature probe in the continous water circuit of the thermosalinograph. |

Daily, both types of data, meteorological and TSG, are sent to the IEO Data Center on land, and kept within archives for each month and vessel. Then, formatting and quality control (QC) procedures are applied for its permanent storage within MEDAR/MedAtlas format (Maillard et al., 1998). MEDAR/MedAtlas is an ASCII autodescriptive format available since the last decade of the 20th century, which merges to facilitate reading, recording, processing and storing oceanographic data. Each





file is provided with a heading where all metadata information is reflected alongside a flag column where the QC labels are specified. Data sampling has different frequency for each type of data. In consequence, final fluxes products are calculated on an hourly frequency. Then, with all the information above gathered, final MEDAR/MedAtlas formatted heat and momentum

fluxes data archives are constructed for each month and vessel. The columns show the three air-sea interaction fluxes, *SHFL*, *LHFL* and *MOFL* along with the variables used in their calculation (Table 1). A flag column is also added based on the criteria specified in the QC section. Besides, incident radiation (*RDIN*) is also included in order to bring the possibility to determine the total heat storage (HS) in the ocean (Eq. 1).

$$HS = Q_s + Q_{lw} + SHFL + LHFL \qquad (1)$$

where $Q_s$ is the short wave incident radiation from the sun, $Q_{lw}$ is the net long wave radiation. *RDIN* accounts for both shortwave and longwave incident radiation from the sun and atmosphere. The upward longwave radiation from the sea can be obtained with Stefan-Boltzman radiation law using the sea temperature (*TEMP*), $R = \epsilon \sigma T^4$ (Stull, 2012); where $\epsilon$ is the emissivity ranging between 0.93 and 1 (Fung et al., 1984) and $\sigma$ is the Stefan-Boltzmann constant. Thus, $Q_s$ and $Q_{lw}$ are addressed by the data set, and alongside the air-sea interaction heat fluxes, both radiative and convective exchange between the

two media are determined.

## 3.2 Validation and QC

In order to obtain reliable data products, it is important to elaborate adequate validation and QC procedures. Both types of records -meteorological and TSG data, as well as subsequent fluxes products- are revised and checked by manual and auto-

matic techniques. As it was previously mentioned, all MEDAR/MedAtlas archives contain a flag column which specifies the QC of each variable and individual data sample. The flag criteria is based on SeaDataNet guidelines (Schaap and Lowry, 2010).

Firstly, meteorological and TSG data are validated by visual recognition, useful to check the reliability of the data set. It enables to understand the noise of the time series, while simultaneously identifying immediate anomalous or spike records. Thus,

the time series of each year data record for each data type is represented, allowing to observe whether all measurements are within the temporal limits specified in the metadata heading. In addition, map-plotting of the vessel's trajectories is a practical procedure to detect position errors. Additionally, other automatic processes are executed. A global range test for each variable is conducted by setting highly extreme thresholds, which would be impossible to exceed in real life; e.g, >100% or negative values for relative humidity. Also, more restrictive limits are fixed taking into account regional and seasonal climatological

behaviours. North Atlantic area does not reach such high sea surface and air temperature values as Western Mediterranean or Canary regions. Further, additional QC tests are applied based on SeaDataNet recommendations (Schaap and Lowry, 2010). A gradient test (GT), Eq. (2), which evaluates if the gradient of a measurement and the previous and next values is too sharp, is applied for each variable,

$$GT = |V_2 - (V_3 + V_1)/2| \qquad (2)$$





where $V_2$ is the record being checked and $V_1$ and $V_3$ the previous and next measurements. If GT is higher than the values specified in Table 2, it is flagged as 4. Temperature data are also tested under a spike test (ST), Eq. 3,

**Table 2.** GT thresholds used for each variable during the QC process.

| Data Type | Variable | GT threshold |
|---|---|---|
| Meteorological | DRYT (°C) | 1.5 |
| | ATMS (hPa) | 1 |
| | WSPD (m/s) | 3 |
| | RELH (%) | 10 |
| TSG | TEMP (°C) | 1 |

$$ST = |V_2 - (V_3 + V_1)/2| - |(V_3 - V_1)/2| \qquad (3)$$

SeaDataNet guidelines fix the ST in 6°C (if ST > 6°C $V_2$ fails the test). However, noticing the large threshold this quantities supposes, it is implemented another fixed limit based on statistical analysis (Paladini de Mendoza et al., 2022). This new thresh-

old is calculated as 3×IQR parameter; IQR defined from the first (Q1) and third (Q3) quantiles of the monthly distribution, IQR=Q3-Q1. The most restrictive threshold, 6°C or 3×IQR, is the reference for each test. Lastly, additional QC procedures are applied: if a constant value is repeated notably during the record, it is flagged as 4; and even if under all the tests applied there exist notorious eye-catching spikes values to remove, they are flagged manually based on the expertise developed throughout the study. Regarding the final fluxes data set, if any of the meteorological or TSG variables involved in their calculation is

flagged with a value other than 1, the air-sea interaction fluxes are automatically flagged in the same way. In addition, fluxes time series and map trajectories are newly checked in order to, firstly, confirm that all data are placed over the sea, and secondly, to identify any prominent spike which are then flagged manually.

     The time period covered by this study is 2011-2023. However, the data record is not a continuous and uninterrupted time

series. Data records are collected based on the availability of the ship's activities. Additionally, technical problems or preventive actions for its good condition might get the experimental equipment off, creating gaps in the record. This, along with the QC procedures, reduces the amount of valid data available (Fig. 2).





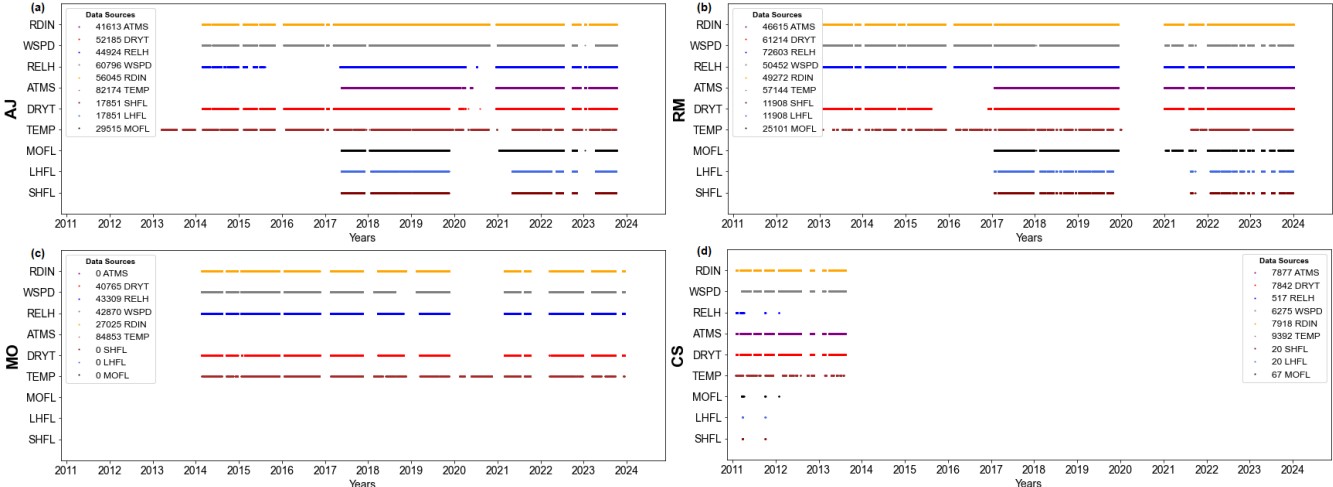

**Figure 2.** Data records timeline from 2011 to 2023 for each meteorological and TSG variables, as well as for *SHFL*, *LHFL*, *MOFL* products for each vessel, AJ (a), RM (b), MO (c), CS (d). Quality control applied. Number of data for each magnitude in the plot legend.

### 3.3 Air-sea interaction fluxes calculation

Air-sea, heat and momentum fluxes are transmitted in the surface layer (SL), the immediate closest layer to the interface be-

tween atmosphere and ocean, and the lowest layer in the ABL (Stull, 2012). Here, the turbulent exchange within both media, its dynamics and implications in the development of the ABL, are ruled by Monin-Obukhov similarity theory (Foken, 2006), which assures that these heat and momentum fluxes are constant with height. Mentioned fluxes are transported by the vertical wind component $\omega$. Hence, sensible and latent heat fluxes can be considered as temperature and humidity fluxes towards the atmosphere, whereas the momentum flux appears as wind stress towards the surface. The experimental measurement of these

quantities requires a temporal resolution enough to capture the turbulent fluctuations, *eddys*, of the magnitudes immersed in these transports. Several experimental approaches exist to determine these quantities: the Eddy Correlation method (Large and Pond, 1982); or the dissipation method (Pond et al., 1971). However, the most useful methodology to calculate mentioned turbulent fluxes, and which is used in this study due to the experimental and technical constrictions, is the bulk aerodynamic approximation (Grachev and Fairall, 1997) (Eq. 4, 5, 6). Here, a relationship between air-sea fluxes and the gradient of tem-

perature, specific humidity, and horizontal wind, respectively, is established; between the sea surface and the corresponding measurement point in the atmosphere

$$SHFL = \rho C_p C_T U(z)(T_s - T_a(z)) \tag{4}$$

$$LHFL = \rho L C_E U(z)(q_s - q_a(z)) \tag{5}$$






$$MOFL = \rho C_D U^2(z) \tag{6}$$

where subindex $s$ makes reference to the sea surface whereas $a$ to the atmosphere; $\rho$ is the air density; $C_p$ and $L$ are the specific heat at constant pressure and the latent heat of evaporation, respectively; $U$ is the wind speed, $T$ and $q$ are the temperature and the specific humidity of the media; and $C_T$, $C_E$ and $C_D$ are aerodynamic bulk coefficients. Following this criteria,

*SHFL* is positive towards the atmosphere if sea temperature is higher than air temperature; whilst *LHFL* is positive when there is evaporation. *MOFL* is always positive due to the wind stress from the atmosphere to the sea surface. In ideal conditions *U(z)* is the wind speed relative to the sea surface current, which can be measured using specific equipment such as scatterometers (Edson et al., 2013; Chacko et al., 2022). In this case, ultrasonic anemometers are used to measure wind speed and direction, unable to determine the sea current speed. Thus, *U(z)* is considered henceforth as the absolute wind velocity neglecting the

effects of the surface ocean current. Previous studies have pointed to errors between 10-20 % when neglecting ocean currents around 0.5 ms$^{-1}$ (Trenberth et al., 1990).

$C_T$ and $C_E$ are the aerodynamic heat and moisture coefficients, respectively. They depend on the stability of the SL as well as the wind speed. Several researches have found values of C$_T$ between $1.1 \times 10^{-3}$ for unstable stratification and $0.75 \times 10^{-3}$

in a stable atmosphere (Smith, 1980). On the contrary, for C$_E$ Anderson and Smith (1981) concludes that it depends more on wind speed than on stability. Hence, for this study, due to the limitation to obtain the state of the stability of the ABL -it would be essential to calculate the characteristic turbulent scales of temperature, wind and moisture according to Monin-Obukhov theory-, the bulk aerodynamic coefficients are considered as constants: $1 \times 10^{-3}$ for C$_T$ (Pond et al., 1971) and $1.3 \times 10^{-3}$ for C$_E$ (Anderson and Smith, 1981). Furthermore, for the drag coefficient, C$_D$, the wind-dependency relation exposed in Large

and Pond 1981, implemented in the python routines Air- Sea, (https://github.com/pyoceans/python-airsea) is used. The air density, $\rho$, is calculated using the ideal gases equation (Iribarne and Godson, 2012). The specific humidity is derived from the pressure and relative humidity measurements. It is calculated using the specific humidity in saturated conditions based on the vapour pressure dependency on temperature according to Buck (1981), which is also addressed by the aforementioned Air-Sea routines; and the relation $RH = q/q_{saturated}$, being $RH$ the relative humidity. The specific humidity, of the sea surface, $q_a$,

is obtained analogously but considering saturation conditions. This last is multiplied by a 0.98 factor due to the reduction of saturated vapour pressure in consequence of salt concentration (Large and Yeager, 2009). Specific heat at constant pressure, $C_p$ has a value of 1004.7 Jkg$^{-1}$K$^{-1}$, whereas latent heat of evaporation $2.5 \times 10^6$ Jkg$^{-1}$.

For these calculations, the disponibility of the meteorological and TSG variables measurements is needed. The absence

of any variable involved in Eq. (4), (5), (6) makes it impossible to determine *SHFL*, *LHFL* and *MOFL*. For the period from 2013 to 2016, there is no pressure measurements in any of the AJ, RM, MO vessels (Fig. 2). This also happens during the rest of the years for MO. Nonetheless, the data are still useful to determine the sign of the fluxes -difference of temperature or

humidity between ocean and atmosphere in Eq. (4), (5)-. Given that, despite the concrete value cannot be obtained, its tendency, proportionality and behaviour can still be inferred.

# 4 Results and discussion

## 4.1 Outcomes availability

Once the QC is applied and the data are in a well-suited MEDAR/MedAtlas format, results and analysis from AJ, RM, MO and CS vessels' measurements can be obtained. During their oceanographic activities, the vessels cover a notorious area near the Spanish waters (Fig. 3a). The total sum of the entire data sampling between 2011 and 2023 is 211713 rows of meteorological and TSG data and their subsequent air-sea fluxes products. However, the measurements flagged with '9' and '4' (meaning no data, and error data, respetively) reduce the operative number of data available for each variable. In Table 3 the amount (and percentage) flagged as 'good value' ('1') for each variable is shown, whereas in Table 4 the flux products flagged as '4' for each marine region is specified. The atmospheric pressure is the variable with less good values (44.6 %), due to the lack of measurement during the period 2013-2016 in AJ, RM and MO vessels, although accounts for a significant amount of 94572 values. It is followed by the sea temperature (62.7 %), which affects directly to the heat fluxes, the reason why there exists such a remarkable difference between the percentage of heat (14.1 %) and momentum (25.8 %) fluxes. The absence of good values of RELH stands out mainly during the CS's trajectories between 2011 and 2013, when values above 100 % and below 0 % that prevent the correct determination of the air density are recorded; although its percentage of 'good values' is a notorious 76.5 %. Also, some slightly disproportionate wind speed measurements recorded by CS are flagged as 4; nonetheless, it presents a high percentage of 'good values'. Furthermore, *DRYT* exceeds these percentages, with 77.5 %, despite presenting occasional high values registered during winter months in 2018 and 2019.

**Table 3.** Number (and percentage) of good values -data flagged as 1- for each variable within the total four vessels' records

| Variables | ATMS | DRYT | WSPD | RELH | TEMP | RDIN | SHFL/LHFL | MOFL |
|---|---|---|---|---|---|---|---|---|
| Good values | 94572 | 164098 | 156301 | 161989 | 132856 | 133727 | 29779 | 54683 |
| Porcentage | 44.6 % | 77.5 % | 73.8 % | 76.5 % | 62.7 % | 63.2 % | 14.1 % | 25.8 % |

**Table 4.** Number of data flagged as 4 in the final flux data products for each region. In addition, the corresponding percentage of the total number of available data is displayed.

| Turblent fluxes | North Atlantic | | South Atlantic | | Canary | | Alboran | | Levantine | | West-Mediterranean | | Cantabrian-Atlantic | | Atlantic | |
|---|---|---|---|---|---|---|---|---|---|---|---|---|---|---|---|---|
| Heat fluxes | 1528 | 13.5 % | 888 | 19.5 % | 370 | 5.6 % | 340 | 18.8 % | 1321 | 20.9 % | 1662 | 20.6 % | 1541 | 12.8 % | 309 | 10.7 % |
| Momentum flux | 3355 | 12.4 % | 1544 | 18.3% | 764 | 7.1 % | 894 | 29.9 % | 2640 | 27.0 % | 3527 | 27.7 % | 3375 | 12.1 % | 679 | 16.1% |

Figure (3b, c, d) are examples of oceanographic visual products that can be represented with this data set using ODV (Schlitzer, 2022), where air-sea fluxes calculated are plotted throughout the vessel's trajectories in the points where they are

available. These plots give a visual idea of the spatial distribution and magnitude of each variable. There can be noticed differences between basins. The Mediterranean area exhibits higher evaporation amounts in comparison with the Atlantic facade. Sensible heat characteristics are less remarkable in these areas. Nevertheless, negative values in the Alboran sea and the Atlantic coast in the north west of Spain stand out. In addition, the effect of the wind stress due to the trade winds is also noticeable in the Canary region for all air-sea interaction fluxes.


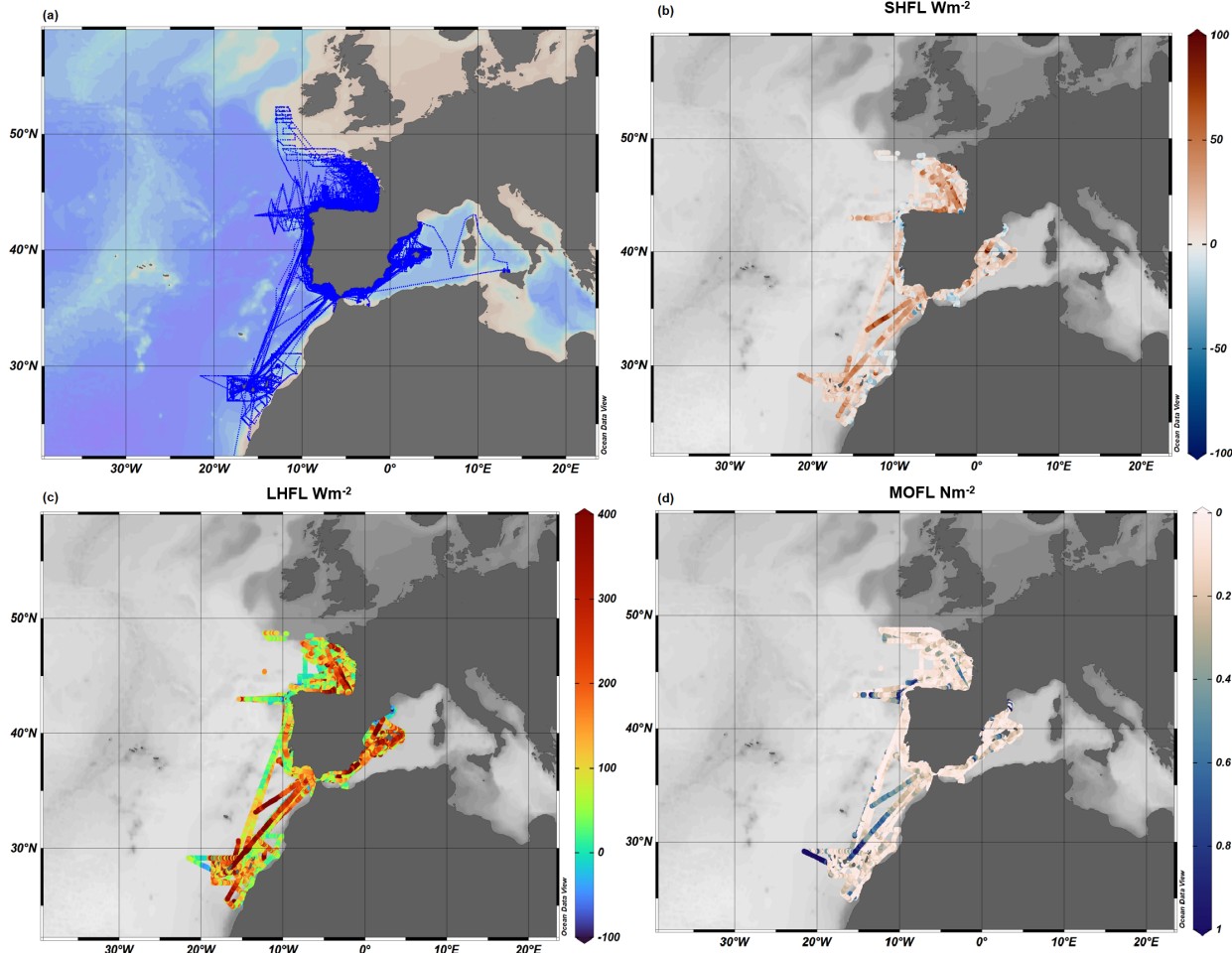

**Figure 3.** *SHFL* (b), *LHFL* (c) and *MOFL* (d) along the AJ, RM, MO, CS vessel's trajectories (a) during the whole period 2011-2023.

## 4.2 Annual cycles

Further analysis processes can be carried on with these data sets. Annual cycles provide key information for understanding the mean behaviour and variability of a field within inter-annual frequencies. They also contribute to both validating the data and identifying key or eye-catching patterns to be further analysed. In this case, Table 5 shows the annual cycle for the sensible heat





flux for each sea region considered. All regions manifest a similar behaviour, with the lowest values during summer season and the highest results for November and December months, showing a greater loss of heat by the ocean during winter months due to the contrast of temperature between the sea surface and air. The annual mean is positive over the whole domain, with highest quantities for the Canary (9.86 Wm$^{-2}$) and Levantine-Balearic (6.71 Wm$^{-2}$) regions, excluding the Alboran Sea where several negative values are present and which conforms an annual mean of -2.26 Wm$^{-2}$. This special behaviour of this region, with an

average annual cooling, in contrast with other basins, is also detailed in the scientific bibliography (Vargas-Yáñez et al., 2007; Ruiz et al., 2008). Some hypotheses to be considered to explain this behaviour are, the income of colder north Atlantic waters, or the upwelling processes conducted by the Poniente wind regime (Ortega et al., 2023) and the two anticyclonic gyres in this basin (Sarhan et al., 2000). The standard deviation values, despite presenting amounts even higher than the mean result, are compatible with other studies.

**Table 5.** *SHFL* annual cycle for each region. Mean and standar deviation (Std) for each month during the period 2011-2023.

| Sensible heat flux (Wm$^{-2}$) | North Atlantic | | South Atlantic | | Canary | | Alboran | | Levantine | | West-Mediterranean | | Cantabric-Atlantic | | Atlantic | |
|---|---|---|---|---|---|---|---|---|---|---|---|---|---|---|---|---|
| Month | Mean | Std | Mean | Std | Mean | Std | Mean | Std | Mean | Std | Mean | Std | Mean | Std | Mean | Std |
| January | 4.85 | 23.26 | - | - | 9.73 | 11.40 | 8.29 | 6.20 | 9.33 | 14.78 | 9.11 | 13.14 | 4.85 | 23.26 | 9.26 | 5.42 |
| February | -0.41 | 16.43 | 10.60 | 18.80 | 5.61 | 12.93 | 6.72 | 15.55 | 15.58 | 16.83 | 12.89 | 17.09 | -0.40 | 16.43 | 11.04 | 17.60 |
| March | 8.11 | 16.57 | 8.94 | 18.11 | 12.49 | 20.36 | -10.70 | 14.61 | -14.14 | 12.37 | -11.67 | 13.34 | 7.85 | 16.14 | 10.50 | 20.08 |
| April | 6.37 | 18.18 | -5.99 | 19.78 | 12.24 | 14.10 | -16.61 | 29.42 | -0.46 | 16.64 | -1.29 | 17.78 | 6.37 | 18.19 | 8.40 | 10.40 |
| May | 4.45 | 12.10 | -0.93 | 15.21 | 11.31 | 9.87 | -12.18 | 15.45 | 8.82 | 20.28 | -1.11 | 19.67 | 4.30 | 12.50 | 13.50 | 15.13 |
| June | 0.28 | 13.02 | 6.42 | 17.05 | 4.57 | 11.30 | -0.61 | 22.21 | 2.14 | 10.19 | 1.96 | 11.45 | 0.28 | 13.02 | 3.93 | 17.97 |
| July | 2.74 | 10.72 | 3.19 | 18.31 | 21.18 | 9.83 | -18.10 | 22.49 | 3.36 | 11.35 | -0.14 | 13.99 | 2.74 | 10.72 | 3.14 | 15.71 |
| August | 5.95 | 17.30 | 2.52 | 12.98 | – | – | -3.44 | 14.54 | 7.17 | 13.50 | 3.30 | 14.96 | 5.05 | 17.36 | -5.61 | 13.90 |
| September | 8.09 | 17.75 | 8.62 | 14.05 | 2.62 | 4.35 | -4.02 | 16.40 | 13.98 | 13.38 | 8.70 | 16.55 | 8.34 | 17.40 | -2.28 | 15.41 |
| October | 8.21 | 16.61 | 7.02 | 17.41 | 5.53 | 12.47 | -4.51 | 10.12 | 16.17 | 16.34 | 11.44 | 17.50 | 9.21 | 16.61 | 7.12 | 22.83 |
| November | 22.40 | 16.31 | 17.86 | 18.00 | 18.64 | 12.50 | 0.61 | 19.58 | 13.25 | 16.30 | 9.71 | 18.115 | 22.40 | 16.31 | 12.41 | 16.01 |
| December | 20.06 | 26.97 | 21.71 | 18.41 | 13.48 | 10.88 | 28.10 | 20.02 | 27.61 | 14.18 | 28.16 | 18.01 | 20.06 | 26.97 | 26.40 | 28.07 |

Regarding latent heat flux (Table 6), the annual cycle pattern is similar to the sensible heat but with greater values: lower quantities during the spring-summer transition, and higher during the winter season when wind stress is more intense, confirming, therefore, the greater loss of heat during winter months due to turbulent exchange. Higher annual mean values are found once again in the Canary (138.35 Wm$^{-2}$) and Levantine-Balearic (139.50 Wm$^{-2}$) regions. These behaviours are expected as the former is located in subtropical latitudes, commonly known to be areas with great levels of evaporation (Hartmann, 2015);

whereas the latter belongs to the Mediterranean basin, where the evaporation regime is equally significant, even exceeding precipitation rates (Talley, 2011). The annual average within the Mediterranean area is slightly superior to other model-based reconstructions (Yu and Weller, 2007; Ruiz et al., 2008); nonetheless, their range of variability are compatible. More moderated values are found within the Cantabrian and Atlantic areas, where precipitation exceeds evaporation. However, there are even significant quantities that must be validated by additional studies.





**Table 6.** *LHFL* annual cycle for each region. Mean and standard deviation (Std) for each month during the period 2011-2023.

| Latent heat flux (Wm$^{-2}$) | North Atlantic | | South Atlantic | | Canary | | Alboran | | Levantine | | West-Mediterranean | | Cantabric-Atlantic | | Atlantic | |
|---|---|---|---|---|---|---|---|---|---|---|---|---|---|---|---|---|
| Month | Mean | Std | Mean | Std | Mean | Std | Mean | Std | Mean | Std | Mean | Std | Mean | Std | Mean | Std |
| January | 129.36 | 92.63 | - | - | 140.33 | 89.80 | 108.38 | 50.47 | 158.58 | 78.37 | 145.79 | 75.72 | 129.37 | 92.63 | 76.15 | 40.77 |
| February | 92.38 | 61.62 | 108.36 | 83.29 | 119.09 | 93.70 | 137.13 | 79.67 | 149.03 | 70.73 | 145.17 | 73.388 | 92.58 | 61.62 | 130.26 | 96.23 |
| March | 132.47 | 85.79 | 123.92 | 91.63 | 144.36 | 96.24 | 236.46 | 49.96 | 165.13 | 66.46 | 200.94 | 70.60 | 128.50 | 85.20 | 141.48 | 106.27 |
| April | 133.00 | 92.45 | 72.09 | 58.63 | 138.78 | 87.51 | 18.02 | 67.01 | 44.85 | 70.96 | 43.49 | 71.05 | 133.04 | 92.45 | 119.53 | 74.19 |
| May | 106.59 | 81.06 | 9.38 | 81.77 | 122.92 | 74.75 | 96.96 | 46.10 | 86.89 | 6107 | 81.48 | 61.70 | 113.06 | 80.98 | 118.46 | 85.77 |
| June | 65.84 | 73.70 | 114.88 | 96.57 | 149.97 | 74.57 | 186.34 | 152.41 | 112.05 | 87.74 | 116.93 | 95.32 | 65.84 | 73.70 | 158.18 | 117.22 |
| July | 102.96 | 85.43 | 115.18 | 100.03 | 201.81 | 67.93 | 60.85 | 120.00 | 114.27 | 86.80 | 105.47 | 95.41 | 102.96 | 85.43 | 122.76 | 63.49 |
| August | 113.15 | 94.53 | 110.02 | 97.56 | – | – | 48.45 | 65.45 | 177.17 | 127.21 | 130.17 | 126.06 | 113.14 | 94.53 | 175.71 | 135.03 |
| September | 136.55 | 101.47 | 151.62 | 101.99 | 62.65 | 40.74 | 167.98 | 221.06 | 226.96 | 141.38 | 201.97 | 168.49 | 133.27 | 98.57 | 93.75 | 81.56 |
| October | 90.78 | 90.15 | 135.47 | 96.88 | 124.27 | 97.93 | 108.37 | 62.50 | 211.88 | 141.28 | 188.83 | 135.05 | 90.78 | 90.15 | 96.50 | 121.97 |
| November | 201.35 | 98.54 | 166.48 | 125.02 | 151.95 | 118.46 | 163.11 | 134.79 | 157.48 | 112.16 | 156.12 | 117.92 | 201.36 | 98.5 | 149.50 | 100.70 |
| December | 124.28 | 84.23 | 155.20 | 87.55 | 173.66 | 105.51 | 212.85 | 91.36 | 173.71 | 85.70 | 203.15 | 87.17 | 124.28 | 84.23 | 184.59 | 139.39 |

Momentum flux outcomes present expected values within the characteristics ranges of the specific location (Table 7). The annual mean values for all regions are around 0.1 Nm$^{-2}$. This quantity is exceeded in several regions during November and December months enabling greater turbulent exchange between the ocean and the atmosphere. This is logical due to the atmospheric baroclinicity and the intensification of the wind regimes of these months. However, annual cycles are not satisfactorily defined for all regions. Alboran-Strait area shows higher-than-expected values during July and August, in contrast with the inter-annual wind climatology of their regional winds: Levante-Poniente regime (Camacho et al., 2022). Canary region also presents higher results during August, although this is already documented in the bibliography (Azorin-Molina et al., 2018).

**Table 7.** *MOFL* annual cycle for each region. Mean and standar deviation (Std) for each month during the period 2011-2023.

| Momentum Flux (Nm$^{-2}$) | North Atlantic | | South Atlantic | | Canary | | Alboran | | Levantine | | West-Mediterranean | | Cantabric-Atlantic | | Atlantic | |
|---|---|---|---|---|---|---|---|---|---|---|---|---|---|---|---|---|
| Month | Mean | Std | Mean | Std | Mean | Std | Mean | Std | Mean | Std | Mean | Std | Mean | Std | Mean | Std |
| January | 0.051 | 0.11 | 0.050 | 0.047 | 0.073 | 0.102 | 0.047 | 0.089 | 0.127 | 0.170 | 0.105 | 0.156 | 0.051 | 0.113 | 0.063 | 0.066 |
| February | 0.035 | 0.060 | 0.174 | 0.105 | 0.088 | 0.146 | 0.157 | 0.241 | 0.086 | 0.088 | 0.103 | 0.146 | 0.036 | 0.060 | 0.113 | 0.097 |
| March | 0.074 | 0.132 | 0.142 | 0.106 | 0.101 | 0.134 | 0.301 | 0.241 | 0.069 | 0.089 | 0.112 | 0.153 | 0.075 | 0.130 | 0.208 | 0.224 |
| April | 0.061 | 0.111 | 0.096 | 0.072 | 0.090 | 0.126 | 0.053 | 0.617 | 0.065 | 0.098 | 0.063 | 0.096 | 0.061 | 0.111 | 0.169 | 0.156 |
| May | 0.080 | 0.120 | 0.137 | 0.091 | 0.084 | 0.112 | 0.142 | 0.223 | 0.100 | 0.151 | 0.107 | 0.171 | 0.082 | 0.117 | 0.157 | 0.142 |
| June | 0.055 | 92.097 | 0.108 | 0.072 | 0.084 | 0.082 | 0.273 | 0.270 | 0.063 | 0.089 | 0.075 | 0.118 | 0.055 | 0.097 | 0.192 | 0.167 |
| July | 0.055 | 0.097 | 0.091 | 0.063 | 0.302 | 0.178 | 0.273 | 0.275 | 0.055 | 0.073 | 0.083 | 0.142 | 0.055 | 0.105 | 0.211 | 0.170 |
| August | 0.074 | 0.106 | 0.075 | 0.060 | – | – | 0.043 | 0.065 | 0.055 | 0.062 | 0.050 | 0.062 | 0.064 | 0.105 | 0.198 | 0.161 |
| September | 0.072 | 0.097 | 0.119 | 0.070 | 0.026 | 0.038 | 0.221 | 0.246 | 0.100 | 0.125 | 0.145 | 0.189 | 0.076 | 0.105 | 0.090 | 0.121 |
| October | 0.080 | 0.116 | 0.104 | 0.064 | 0.062 | 0.089 | 0.063 | 0.084 | 0.072 | 0.086 | 0.070 | 0.086 | 0.081 | 0.101 | 0.080 | 0.103 |
| November | 0.041 | 0.074 | 0.104 | 0.066 | 0.065 | 0.136 | 0.181 | 0.231 | 0.058 | 0.093 | 0.087 | 0.148 | 0.042 | 0.074 | 0.095 | 0.106 |
| December | 0.035 | 0.093 | 0.164 | 0.131 | 0.113 | 0.156 | 0.128 | 0.122 | 0.062 | 0.056 | 0.121 | 0.142 | 0.035 | 0.094 | 0.1567 | 0.160 |





## 4.3 Time series

The time series of heat and momentum fluxes are also plotted to analyse the evolution, tendencies and gaps during the period
considered. The Western Mediterranean and the Cantabrian-Atlantic graphics are shown in Fig. 4 as they are the regions with
more extension in this study. The data gap during the year 2020 and the beginning of 2021, due to the difficult working con-
text of the COVID pandemic, stands out. Uniquely, the AJ's data are available during this time frame, although with several
problems in the temperature probe and the barometer that makes it impossible to make a reliable calculation of the final prod-
ucts. These gaps are difficult to interpolate due to the high variability of the meteorological magnitudes implied in the fluxes
calculation, creating a complicated context to study hypothetical tendencies or petterns, which are blurred by the intermittence
of the time series. A thorough maintenance and care to the measuring equipment is fundamental in order to take advantage
of the experimental measuring potential. However, several important features can be subtracted: the Mediterranean area ex-
hibit greater latent heat fluxes due to the dominance of evaporation in its basin. Sensible heat is higher for Cantabrian basin;
the negative values from the Alboran sea, aforementioned, decreases the mean value of the Mediterranean basin. Regarding
momentum flux, mean values for all the basins are around 0.1 $\mathrm{Nm^{-2}}$, which is an expected quantity for these basins (Samuel
et al., 1999; Trenberth et al., 1990), with the highest value for the Alboran Sea, presumably due to the Levante-Poniente wind
regime influence (Ortega et al., 2023).

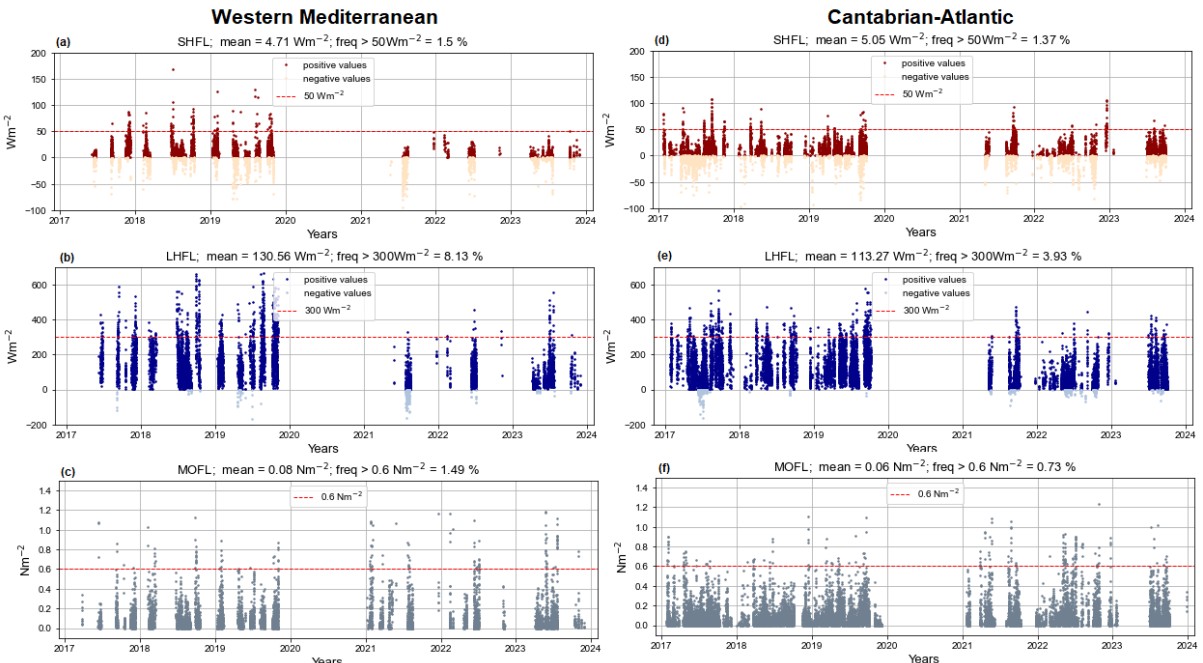

**Figure 4.** *SHFL*, *LHFL* and *MOFL* time series (2017-2023) for Western-Mediterranean (a,b,c) and Cantabrian-Atlantic (d,e,f) regions. Red dashed line points the threshold of 50 $\mathrm{Wm^{-2}}$, 300 $\mathrm{Wm^{-2}}$ and 0.6 $\mathrm{Nm^{-2}}$ respectively for each magnitude.



The level of 50 Wm$^{-2}$ and 300 Wm$^{-2}$ for sensible and latent heat respectively, which are considered significant magnitudes for both variables, are marked in red dashed lines. The regions of the planet where the greatest turbulent heat exchange between

the ocean and the atmosphere occurs are the western margins of the Atlantic and Pacific oceans, in the Gulf and Kuroshio currents respectively, where these thresholds are exceeded (Hartmann, 2015). On the other hand, 0.6 Nm$^{-2}$ threshold is defined for momentum flux, based on the results of other studies in open ocean conditions (Large and Pond, 1981). Regions with remarkable wind intensity regimes, such as the Southern Ocean, can reach values superior to 1 Nm$^{-2}$ (Trenberth et al., 1990; Morrow et al., 1992); or even greater (2-8 Nm$^{-2}$) during tropical cyclones (Morrow et al., 1992). The Alboran sea expresses

one of the highest frequencies (0.76%) of momentum flux exceeding this threshold, while it has the lowest ratio above the 50 Wm$^{-2}$ sensible heat threshold -also expected because of its negative value average-. In addition, the Levantine area shows the highest frequency (8.80 %) above the 300 Wm$^{-2}$ barrier. On the contrary, the Cantabrian region has the lowest ratio for this variable (3.93 %), indicating a lower evaporation rate in comparison.

Furthermore, in order to address the issue of the lack of heat and momentum flux outcomes due to the absence of a necessary variable for their calculation, the evolution of key variables that significantly influence the final products -Eq. (4), (5), (6)- can be examined separately (Fig. 5). The temperature difference is directly proportional to the sensible heat flux, and it allows to define the sign of this flux and assess whether it implies a positive or negative heat contribution to ocean warming. Additionally, wind speed is proportional to all three air-sea interaction fluxes, so its behaviour is crucial for inferring the magnitude of these

products. Besides, relative humidity is essential for determining the specific humidity of the air, and thus the moisture difference with the ocean, which is proportional to latent heat flux. The higher the relative humidity, the less likely water evaporation from the ocean will occur, as the environment reaches saturation; in consequence, this potential energy remains stored in the ocean rather than being transferred to the atmosphere.





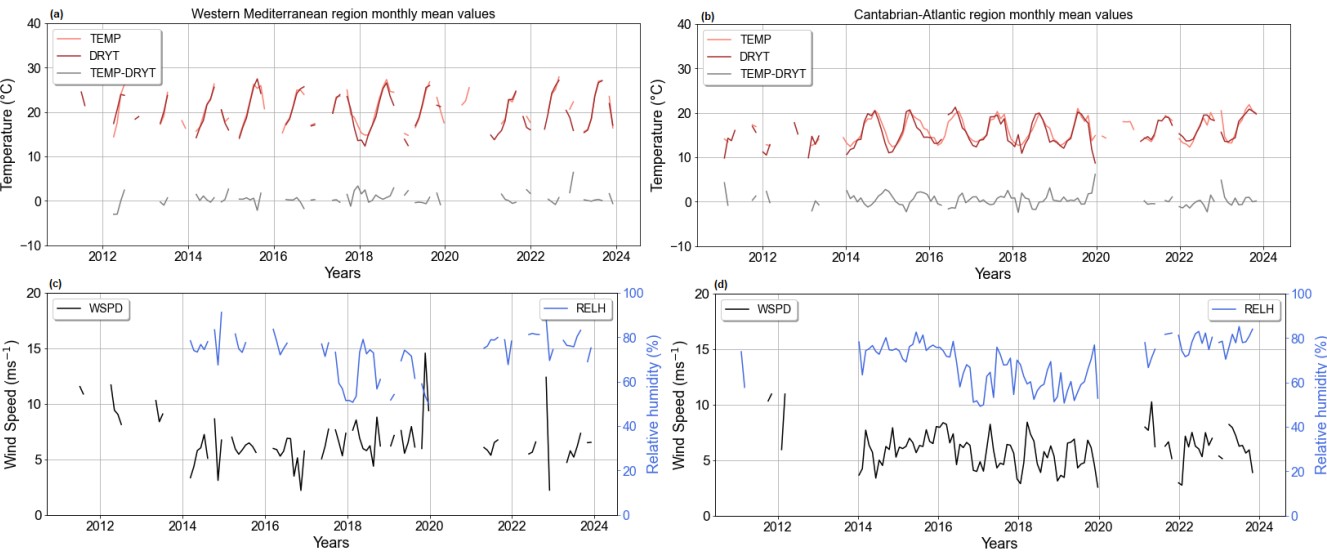

**Figure 5.** Time series of monthly annual variables: sea temperature (orange), air temperature (red) and its difference in grey (a, b); and wind speed (c,d) for Western-Mediterranean (first column) and Cantabrian-Atlantic regions (second column).

## 4.4 Heat budget

One additional application of the study of air-sea interaction fluxes is to obtain the net heat storage balance. Ocean heat budget is determined by both radiative (net shortwave and longwave radiation) and convective components. Sensible and latent heats account for the convective contribution, whereas the incident radiation is provided by *RDIN* variable (Fig. 6c, f) measured by the meteorological measuring equipment. It accounts for the solar (shortwave) and atmospheric (longwave) radiation. The longwave emission by the ocean can be determined by the Stefan-Boltzman law, with the specific determination of the em-

misivity constant for ocean water (Fung et al., 1984). Beyond, given that the ocean it is not a fixed water mass, advective heat term might have an important magnitude to be considered in the heat balance (Anderson, 1952). The Atlantic Meridional Overturning Circulation (AMOC) accounts for the most of the northward transport of heat by the mid-latitudes northern hemisphere (Trenberth and Fasullo, 2017). It reaches its peak heat transport around 26°N; where, consequently, the RAPID Climate Change Programme was established in 2004 to provide direct observational measurements of its flow and transport (Johns

et al., 2011). Thus, this certainly has an important impact on the heat storage balance of the Canary region, which is close to this latitude, not to mention on the rest of the atlantic marine regions considered in this study. Regarding to the Mediterranean sea, mooring based observations have established the net heat transport from the Atlantic through the Strait of Gibraltar to 5.2 $\mathrm{Wm^{-2}}$ (Macdonald et al., 1994). Other authors have determined this flux between 8.5 $\mathrm{Wm^{-2}}$ (Bethoux, 1979) and 5 $\mathrm{Wm^{-2}}$ (Bunker et al., 1982) based on measurements of the net water inflow and its temperature.


The Fig. 6 shows the annual cycles of the heat convective fluxes (a,d; b,e) and the incident radiation (c,f) for two example regions, the Western Mediterranean and the Cantabrian-Atlantic areas. Sensible and latent fluxes reach its minimum strength

during the spring-summer transition, whereas they are more prominent during the colder seasons. On the contrary, the incident radiation presents a reverse pattern. During summer, less heat is lost by turbulent processes and more acquired by radiation, increasing the total heat budget. On the other hand, in winter, when there is a positive and marked contrast between the sea and air temperature, sufficient wind stress to impulse the convective fluxes, and less incident radiation, ocean experiences a decrease in its heat storage.

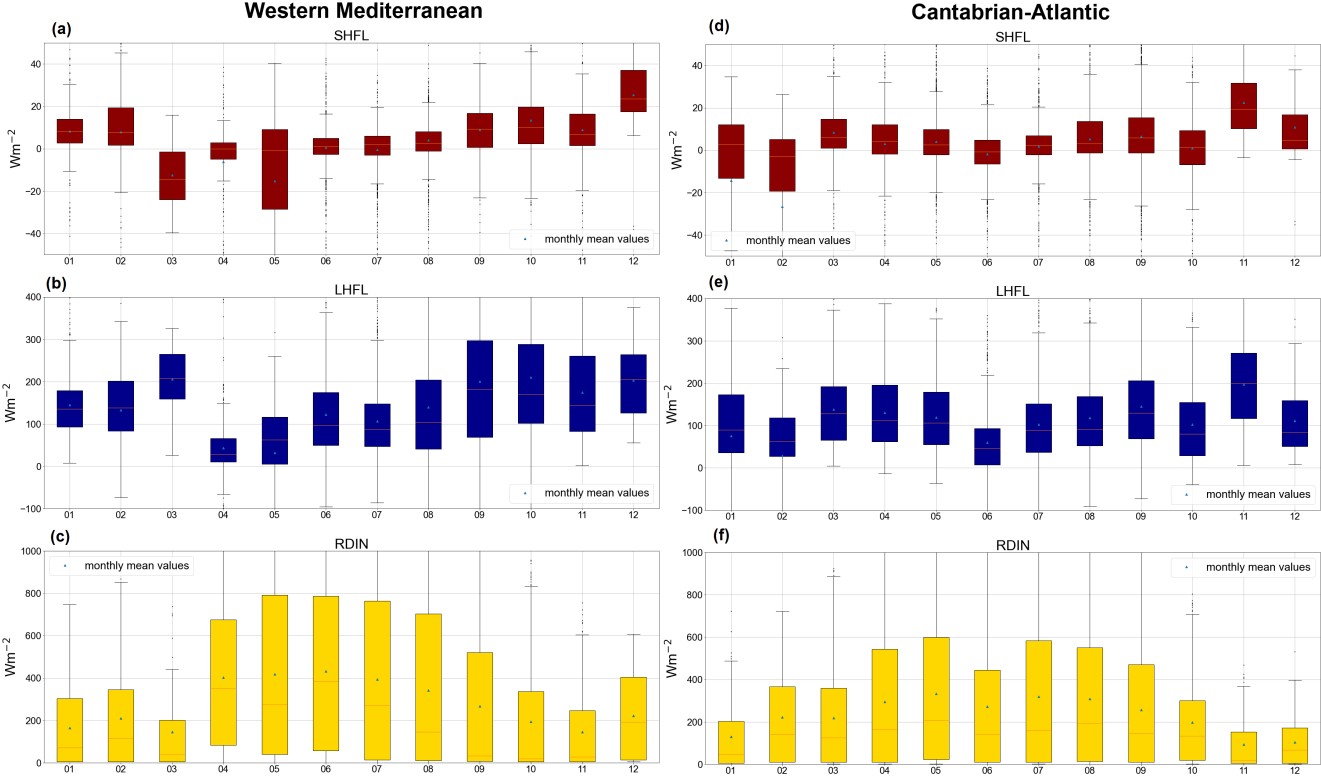

**Figure 6.** Annual cycle of *SHFL* (a), *LHFL* (b), and total incident radiation (RDIN,c) during the period 2011-2023 for the Western-Mediterranean (first column) and Cantabrian-Atlantic (second column) areas.

Incident radiation is greater for the Mediterranean region than for the Cantabrian, due to their differences in geographical location and cloud cover. However, in comparison to other studies in the former area (Lozano et al., 2023), monthly mean radiation values might be presenting some deficits, which contributes to underestimate the heat income stored in the ocean, altering the total heat balance. Further comparisons and validation procedures must be done regarding incident radiation in order to obtain a fairly reliable heat storage outcomes.



## 5 Data availability

All data are publicly available at the SEANOE (SEA scieNtific Open data Edition) service of SeaDataNet. The data have been
classified by vessel and month within MEDAR/Medatlas format. The data set DOI for each research vessel are shown in Table
8.

**Table 8.** DOIs for each data set for each research vessel publicly available at SEANOE.

| Research Vessel | DOI | Reference |
|---|---|---|
| AJ | https://doi.org/10.17882/103856 | (Sánchez-Lorente and Tel, 2024a) |
| CS | https://doi.org/10.17882/103424 | (Sánchez-Lorente and Tel, 2024b) |
| MO | https://doi.org/10.17882/103903 | (Sánchez-Lorente and Tel, 2024c) |
| RM | https://doi.org/10.17882/103855 | (Sánchez-Lorente and Tel, 2024d) |

## 6 Conclusion

This paper presents the air-sea interaction heat and momentum fluxes data obtained through meteorological and TSG obser-
vational measurements collected during the activities of AJ, RM, MO and CS research vessels between 2011 and 2023. The
calculation methodology of the sensible, latent and momentum is the bulk aerodynamic approximation (Large and Yeager,
2009), which relies on the gradient of temperature, specific humidity and horizontal wind between air and sea surface. The
vessels' trajectories cover large part of the Spanish waters and adjacent seas; waters showing notorious heterogeneity due to
the different latitudinal extent, as well as the distribution of the surroundings continental platforms. Thus, this heterogeneous
character encourages the subdivision in different regions proposed in this study.


The hourly sample frequency of the data allows to establish a significant data coverage. This amount of information is re-
vised and checked by quality control procedures so as to obtain reliable and operable data products. The atmospheric pressure
is the variable whit less 'good value' -flagged as 1- results. It presents a percentage of 44.6 % valid values for fluxes calculation
with respect to the total sampling -specially due to the absence of this measurement from 2013 to 2016-, accounting for a
total of 94572 'good values' that are still worth preserving. Other atmospheric variables present better percentages. Relative
humidity and wind speed show considerable notorious percentages, 76.5 % and 73.8 % respectively, despite technicals issues
such as relative humidity values below 0 % and above 100 %, or slightly disproportionate wind velocity; both within the CS
records. On the other hand, air temperature, surpasses these numbers of 'good values', with a percentage of 77.5 % in spite
of the occasional high values recorded by AJ, specifically during 2018 and 2019 winter seasons. To all the concrete technical
issues, the difficult context of the COVID pandemic is added. All in conjunction create gaps in the final fluxes products. This
complicates characterising tendencies; however, behaviours or biases can still be inferred. Thus, it is important to highlight
the need of thorough and rigorous care and maintenance of the vessels' measuring equipment, so as to profit the considerable



investment in the much-needed experimental recording.

Despite the mentioned technical difficulties the values of the three fluxes are within the expected ranges for each region and season. Lower values are found during the summer season, whereas higher ones are present in winter months. Sensible flux present higher values in the Canary region, with an annual mean of 9.86 Wm$^{-2}$; on the contrary, Alboran-Strait area show a negative average of -2.26 Wm$^{-2}$, compatible with other studies (Vargas-Yáñez et al., 2007; Ruiz et al., 2008). In the same line, latent heat flux presents higher values in the Canary and the Levantine-Balearic regions (annual mean of 139.01 and 139.50
Wm$^{-2}$, respectively), although slightly superior values to other model reconstructions (Yu and Weller, 2007; Ruiz et al., 2008). Momentum flux exhibits averages surrounding 0.1 Nm$^{-2}$ for all regions, with larger values within the Alboran-Strait region (annual mean of 0.12 Nm$^{-2}$), influenced by the regional Levante-Poniente wind regime in this area (Camacho et al., 2022). Furthermore, this data set brings the opportunity to study the heat budget storage of the ocean. The total incident radiation is added to the air-sea heat fluxes. The longwave emission from the sea can be obtained using the sea temperature and the
Stefan-Boltzman law with the correct determination of the emmisivity for ocean water (Fung et al., 1984). The total incident radiation exhibits some deficit in different areas, which, added to the slightly high values of the heat fluxes, might establish negative biases to consider within the heat storage balance.

     Experimental records, despite presenting several drawbacks as data-sampling density, incomplete data series or technical
recording issues, are needed for the calibration and validation of other model or satellite-based products that enable a better global coverage and data homogenisation, although, rely necessarily on the observations. Thus, attention must not be lost to this direct information, specially within the ocean. Here, the meteorological and sea surface information, as well as the final fluxes products of the four vessels up to 2023 are presented, nevertheless, AJ, RM, MO are still active research vessels whose recordings can be further used to the same purpose this paper aims, not to mention other research ships of the IEO.

*Author contributions.* ET supervised the project and provided the raw data. ASL prepared the figures and code, and wrote the paper. ASL prepared the datasets. All co-authors reviewed the paper.

*Competing interests.* The authors declare that they have no conflict of interest.

*Disclaimer.* The data set published in this paper is derived from experimental records, presenting the difficulties and drawbacks of field measurements. This records are then used to calculate the final fluxes products using several approximations mentioned within the text. The
reliability and reuse of these methods must be carefully considered by each stakeholder.





*Acknowledgements.* The authors extend their sincere gratitude to the captains and crew of the *Ángeles Alvariño*, *Ramón Margalef*, *Miguel Oliver* and *Cornide de Saavedra* research vessels, whose efforts made possible the data recording. Special acknowledgments are presented to the *Secretaría General de Pesca* of Spain for providing the logistical infrastructure that facilitated the data collection aboard the *Miguel Oliver* vessel. The authors also express profound thanks to the *Instituto Español de Oceanografía*, which enabled the execution of this project within the framework of *JAE INTRO-ICU 2023* fellowship (CSIC).






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
