# Peer review of "Air-sea interaction heat and momentum fluxes based on vessel's experimental observations over Spanish waters"

_Earth System Science Data, 2024_

## Referee Comment (RC2)

**General comments**

The paper analyzes air-sea interaction fluxes—latent and sensible heat (LHFL, SHFL) and momentum (MOFL)—which are key for understanding the ocean-atmosphere interaction and variability. Data were collected from 2011 to 2023 aboard four Spanish research vessels across Spanish and adjacent seas. The data processed and quality-checked by the IEO Data Center, are used to compute the heat and momentum fluxes via bulk aerodynamic methods and stored in MEDAR/MedAtlas format. Regional subdivisions of Spanish waters are used to study airs-sea fluxes behavior, and results are presented. The dataset is publicly available on SeaDataNet/SEANOE data citing and publishing service.

The paper can be accepted for publishing. Below are some issues to be addressed before publication.

**Specific comments**
1) Based on journal rules: *"… for the final accepted publication, a functional data set DOI and its in-text citation must be given in the abstract. If multiple data set DOIs are necessary, please instead refer to the data availability section".* Therefore, authors could add at the abstract (line 19, after SEANOE), the 4 DOIs or make a reference at section 5. To my knowledge, the ESSD publications include the DOIs in the abstract, so I would recommend this option.
2) Line 16: use capital initials (Atmospheric Boundary Layer (ABS)) as for other terms
3) Line 86: where do the 3 additional regions come from? Are there references (as the MSFD regions)? Are the regions mentioned in the Introduction (line 50)? Or they cover the measured area of this work? The linkage is not clear.
4) Table 1: for consistency with the rest of the paper (text, tables, etc.) use as name for the 2$^{nd}$ column the "Variable" instead of "Magnitude" (which is more appropriate).
5) Line 100: besides the reference to the MEDAR/MedAtlas format, authors could also add what standards and controlled vocabularies are used in this format for the (meta)data descriptions such as the P09 (variables names), SeaDataNet flag scale (L20), ICES platform codes (C17) for R/Vs, Instruments (L22), etc. The addition of the general web link would be useful to the reader (https://vocab.seadatanet.org/search).
6) Line 105: The sentence is not clear to me, I do not see any SHFL, LFL, MOFL in Table 1.
7) Line 122: Instead of "visual recognition" I would prefer "visual inspection".
8) Line 122-131: It would be useful for readers to include a Table with the global and regional (and seasonal) ranges for TSG data as it is done below for the Gradient Test thresholds.
9) Line 143: what flag value is used when spikes are detected? Is it "3" according to the SeaDataNet flag scale? It should be mentioned as it is done for flag=4 above.
10) Are specific tools used for the automatic QC checks (gradient, spikes checks) or "in-house" software? As I reader and data manager I am interested to know.
11) Figure 2 caption: without being a native English speaker, I think the last 2 sentences are not correct and can be rephrased to "Data quality control has been applied. Number of data points for each variable is indicated in the plot legend" or "Quality controlled data records timeline …. Number of data points for each variable is indicated in the plot legend".
12) Figure 3, sub-caption (a): I would suggest to add the "Vessels' trajectories" for consistency with the others sub-captions.

13) Line 212: In the section 3.2 of validation and QC, it is mentioned the spikes controls. Are statistics for this QC result available? If yes, these could be included. It should be clear if values with flag=3 are used in the analysis or rejected.

14) Table 3: correct the "Porcentage".

15) Section 4.2: the annual means are computed by averaging the monthly means or by averaging all available data values?

16) Table 5, 7: correct the "standar".

17) Line 329: change the "SEANOE (SEA scieNtific Open data Edition) service of SeaDataNet" to "SEANOE (SEA scieNtific Open data Edition) data citing and publishing service of SeaDataNet". Authors can also add a link here (https://www.seadatanet.org/Software/SEANOE).

18) Line 343: correct the "wit"

19) Line 357: change "present" to "presents".

**Text editing**

20) Check inconsistencies between lines for example: a) line 108 ends with a dot and the equation (1) follows, while line 133 ends with a comma before the equation, b) line 78: at the end of the sentence there is a minus before the dot for example "…2011)-." that should be deleted. Check the paper for other points like (a), (b) and correct.

**References**

21) Most of the references do not include the DOIs. It would be helpful if authors could add them where available (recent publications) so as readers can get quick access at the linked resources.

22) Line 464: change it as: Schlitzer, R.: Ocean data view, https://odv.awi.de, 2025. Also change the reference in the text (line 224).

23) Line476, 478, the two references not in chronological ascending order as the others.

---

## Author Comment (AC1)

**Authors' response to RC2 comments.**

The authors would like to extend a sincere gratitude for the consideration of the suitability of the paper to be accepted for publication. In addition, authors want to express thanks for the attention and the dedication to seeking the highlighted issues mentioned in the comments, which, once corrected, will improve the quality of the text. In the following sections, the correction and discussion of these issues are shown.

**Specific comments**

**1)**

- *RC2:* *''Based on journal rules: ''... for the final acceptd publication, a functional data set DOI and its in-text citation must be given in the abstract. If multiple data set DOIs are necessary, please instead refer to the data availability section.'' Therefore, author could add at the abstract (line 19, after SEANOE), the 4 DOIs or make a reference at section5. To my knowledge, the ESSD publications include the DOIs in the abstract, so I would recommend this option''.*

- *AC:* Data set DOIs added following the comment recommendation.

- *Changes made:* *''The data set generated and described here are publicly available at SEANOE''*, changed to *''The data sets generated and described here are publicly available at SEANOE (https://doi.org/10.17882/103856, https://doi.org/10.17882/103424, https://doi.org/10.17882/10103903, https://doi.org/10.17882/103855)''*

**2)**

- *RC2:* *''Line 16: use capital initial (Atmospheric Boundary Layer (ABS)) as for other terms.''*

- *AC:* Authors agreed.

- *Changes made:* *"atmospheric boundary layer"* changed to *''Atmospheric Boundary Layer'' (ABS)* in line 16.

**3)**

- *RC2:* *''Line 86: where do the 3 additional regions come from? Are there references (as the MSFD regions)? Are the regions mentioned in the Introduction (line 50)? Or they cover the measured area of this work? The linkage is not clear.''*

- *AC:* The three additional regions mentioned in line 87 are three regions that cover the measured area of this work. They are not regions specifically contemplated within the Marine Strategy Framework Directive framework as Spanish marine demarcations, but they are bigger regions where some of these Spanish jurisdictional waters are embedded (and also, where there are a great amount of data to be considered and studied )

**4)**

- *RC2:* *''Table 1: for consistency with the rest of the paper (text, tables, etc.) use as name for the 2nd column the ''Variable'' instead of ''Magnitude'' (which is more appropriate).''*

- *AC:* Authors agree.

- *Changes made:* *''Magnitude''* changed to *''Variable''* in Table 1 .

**5)**

- ***RC2:*** *''Line 100: besides the reference to the MEDAR/MedAtlas format, authors could also add what standards and controlled vocabularies are used in this format for the (meta)data descriptions such as th P09 (variables names), SeaDataNet flag scale (L20), ICES platform codes (C17) for R/Vs, Instruments (L22), etc. The addition of the general web link would be useful to the reader (https://vocab.seadatanet.org/search).''*

- ***AC:*** Following the recommendation, the general web link of the BODC vocabulary library has been added to the text linking it with the variables of the Table 1.

- ***Changes made:*** In line 96, *''The names of the variables and their description follow the standards of the Medatlas Parameter Usage Vocabulary from the British Oceanographic Data Centre (BODC), which can be further explored in https://vocab.seadatanet.org/search''.* Also, in line 107: *''Then, with all the information above gathered, final MEDAR/MedAtlas formatted heat and momentum fluxes data archives are constructed for each month and vessel''* has been changed to *''Then, with all the information above gathered, final MEDAR/MedAtlas formatted heat and momentum fluxes data archives are constructed (following also BODC vocabulary standards) for each month and vessel''*

**6)**

- ***RC2:*** *''Line 105: The sentence is not clear to me, I do not see any SHFL, LFL, MOFL in Table 1.''*

- ***AC:*** Mentioning SHFL, LHFL and MOFL refers to the MEDAR/Medatlat formatted data sets, and Table 1 refers to *''the variables used in their calculation''*. However, following the comment, lines 106-107 have been rewritten for more clarity

- ***Changes made:*** *'' Then, with all the information above gathered, final MEDAR/MedAtlas formatted heat and momentum fluxes data archives are constructed for each month and vessel. The columns show the three air-sea interaction fluxes, SHFL, LHFL and MOFL along with the variables used in their calculation ( Table 1).''* changed to *''Then, with all the information above gathered, final MEDAR/MedAtlas formatted heat and momentum fluxes data archives are constructed for each month and vessel (Sánchez-Lorente and Tel, 2024a, b, c, d). The columns in thesearchive show the three air-sea interaction fluxes, SHFL, LHFL and MOFL along with the variables used in their calculation (which are shown in Table 1).''*

**7)**

- ***RC2:*** *''Line 12: Instead of ''visual recognition'' I would prefer '' visual inspection''. ''*
- ***AC:*** Authors agree.
- ***Changes made:*** Following the recommendation ''visual recognition'' changed to ''visual inspection'' in line 126.

**8)**

- ***RC2:*** *''Line 122-131: It would be usefu for readers to include a Table with the global and regional (and seasonal) ranges for TSG data as it is done below for the Gradient Test thresholds.''*

- ***AC:*** It is an interesting recommendation. However, it has not been decided to add a Table with these ranges because the majority of them would seem logical and/or redundant. For instance, for impossible values such as negative sea surface temperature or relative humidity, it has been decided that a Table does not imply a better comprehension. Also, at regional

scales, for example, the 5-30ºC range has been often chosen as a coherent interval for sea surface temperature (which is not a risky breakthrough), although with little differences or exceptions for different regions that rely on the expertise developed throughout the analysis, and which would be difficult to sum up rigorously in a threshold Table. A similar philosophy is followed with the remaining variables, with thresholds around the 975 hPa and 1030hPa for sea level pressure, being more or less flexible depending on the concrete region considered, but not with unified criteria to resume in a threshold range.

**9)**
- ***RC2:*** *''Line 143: what flag values is used when spikes are detected? Is it ''3'' according to the SeaDataNet flag scale? It should be mentioned as it is done for flag=4 above.''*

- ***AC:*** Spikes detected are flagged as 4, as bad values.

- ***Changes made:*** Line 146 rewritten, '' even if under all the tests applied there exist notorious eye-catching spikes values to remove, they are also flagged manually as 4''.

**10)**
- ***RC2:*** *''Are specific tools used for the automatic QC cheeks (gradient, spikes check) or ''in-house'' software? As I reader and data manager I am interested to know.''*

- ***AC:*** QC checks are done using ''in-house'' developed python code applying the thresholds and tests explained within the text.

**11)**
- ***RC2:*** *''Figure 2 caption: without being a native English speaker, I think the last 2 sentences are not correct and can be rephrased to ''Data quality control has been applied. Number of data points for each variable is indicated in the pol legend'' or ''Quality controlled data records timeline…. Number of data points for each variable is indicated in the plot legend''.''*

- ***AC:*** Authors agree, new Figure 2 caption.

- ***Changes made:*** New Figure 2 caption: ''Data records timeline from 2011 to 2023 for each meteorological and TSG variables, as well as for SHFL, LHFL, MOFL products for each vessel, AJ (a), RM (b), MO (c), CS (d). Quality control applied. Number of data for each magnitude in the plot legend'' changed to '' Quality controlled data records timeline from 2011 to 2023 for each meteorological and TSG variables, as well as for SHFL, LHFL, MOFL products for each vessel, AJ (a), RM (b), MO (c), CS (d). The size of the data sample for each variable is indicated in the plot legend.''.

**12)**
- ***RC2:*** *''Figure 3, sub-caption (a): I would suggest to add the ''Vessel's trajectories'' for consistency ith othe sub-captions.''*

- ***AC:*** Authors agree, new Figure 3 caption.

- ***Changes made:*** New Figure 3 caption: ''SHFL (b), LHFL (c) and MOFL (d) along the AJ, RM, MO, CS vessel's trajectories (a) during the whole period 2011-2023. '' changed to ''AJ, RM, MO and CS vessels's trajectories (a), and SHFL (b), LHFL (c) and MOFL (d) obtained during the extended period 2011-2023''

**13)**

- **RC2:** *''Line 212: In section section 3.2 of validation and QC, it is mentioned the spikes controls. Are statistics for this QC result available? If yes, these could be included. It should be clear if values with flag=3 are used in the analysis or rejected.''*

- **AC:** Only data with flag = 1 (good values) are taken into account in the analysis. Statistics about bad values, flagged with 4., in the final flux products are computed and shown in Table 4.

**14)**
- **RC2:** *''Table 3: correct the ''Porcentage''. ''*

- **AC:** Authors agree.

- **Changes made:** Table 3: ''Porcentage'' changed for ''Percentage''.

**15)**
- **RC2:** *''Section 4.: the annual means are computed by averaging the monthly means or by averaging all available data values?''*

- **AC:** Section 4.2: The annual means are computed by averaging all available data values. Not all months have the same number of measurements, first due to the different number of days in each month, but most importantly, due to the differences in the data sample of each month. Not all days of each month have measurements as the data record is not a continuous and uninterrupted time series, due to the availability of the ship's activities and other technical problems that might take the experimental equipment off. Thus, if the annual average were computed using the monthly averages, it would be overweighting the average of some months against the rest. This is why it was chosen to take into account all the data available, despite the risk of a noisier time series. This question will be reconsidered for future work.

**16)**
- **RC2:** *''Table 5, 7: correct the ''standar''.''*

- **AC:** Author agree.

- **Changes made:** ''standar'' is changed to ''standard'' in Table 5 and 7.

**17)**
- **RC2:** *''Line 329: change the ''SEANOE (SEA scieNtific Open data Edition) service of SeaDataNet'' to ''SEANOE (SEA scieNtific Open data Edition) data citing and publishing service of SeaDataNet''. Authors can also add a link here (https://www.seadatanet.org/Software/SEANOE).''*

- **AC:** Authors agree.

- **Changes made:** Line 331. Following the recommendation, *''SEANOE (SEA scieNtific Open data Edition) service of SeaDataNet''*, is changed to *''SEANOE (SEA scieNtific Open data Edition) data citing and publishing service of SeaDataNet''.*

**18)**
- **RC2:** *''Line 343; correct the ''wit''.''*

- **AC:** Authors agree.

- **Changes made:** Line 347, *''whit''* changed to *''with''*.

**19)**

- **RC2: ''Line 357: change ''present'' to ''presents''.''**

- **AC:** Authors agree.

- **Changes made:** Line 357: *''present''* changed for *''presents''*.

**Text editing section**

**20)**

- **RC2:** ''Check inconsistencies between lines for example: a) line 108 ends with a dot and the equation (1) follows, white line 133 ends with a comma before the equation, b) line 78: at the end of the sentence there is a minus before the dot for example ''...2011)-.'' that should be delated. Check the paper for other points like (a), (b) and correct.''

- **AC:** Homogenized all the equations with a comma before the formula, and all the parentheses using (), not dashes.

**References section**

**21)**

- **RC2:** ''Most of the references do not include the DOIs. It wold be helpful if authors coud add them where available (recent publications) so as readers can get quick access at the linked resources.''

- **AC:** DOIs have been added to the references when available.

**22)**

- **RC2:** ''Line 464: change it as: Schlitzer, R.: Ocean data view, https://odv.awi.de, 2025. Also change the reference in the txt (line 224)''

- **AC:** Authors agree.

- **Changes made:** In line 476, changed ''Schlitzer, R.: Ocean data view, 2022'' to ''Schlitzer, R.: Ocean data view, https://odv.awi.de, 2025''. Also changed ''(Schlitzer, 2022)'' to ''(Schlitzer, 2025)'' in line 227.

**23)**

- **RC2:** ''Line 476, 478, the two reference not in chronological ascending order as the others''.

- **AC:** The reference *''Trenberth, K. E., & Fasullo, J. T. (2017). Atlantic meridional heat transports computed from balancing Earth's energy locally. Geophysical Research Letters, 44(4), 1919-1927.''* has been changed to *''Talley, L. D.: Shallow, intermediate, and deep overturning components of the global heat budget, Journal of Physical oceanography, 33, 530–560, https://doi.org/10.1175/1520-0485(2003)033<0530:SIADOC>2.0.CO;2, 2003.''*, so the previous issue is not a problem anymore.

---

## Author Comment (AC2)

**Authors' response to RC1 comments**

The authors would like to express their sincere gratitude for the interest shown in the submitted manuscript. The following section provides a detailed discussion of the comments raised.

**Major comments**

**1)**
- **RC1:** *''Which monthes are "summer season" refers to?''*

- **AC:** Summer season refers to the months from June to August.

- **Change made:** Line 240: Added the parenthesis *''(from June to August)''*

**2)**
- **RC1**: *''Line 235: Please compare the influence of U(Z) and air-sea temperature difference.''*

- **AC:** This target is out of the scope of the purpose of this data description paper. The idea of this work is to present air-sea heat fluxes data derived from observational measurements and to reflect the potential and the applicability of these data. In this section 4, the annual cycles of all three products are presented, and an explanation of their behaviour is shown linked with the disparities of the difference of temperature between air and sea, or the wind climatology. A comparison between these two factors would comprise a whole unique work of disentanglement of each forcing and attribution to each source, which is not the goal of this paper.

**3)**
- **RC2:** *''Are the thresholds selected for "Validation and QC" GT and ST applicable to the study area of this paper?''*

- **AC:** These GT and ST thresholds are considered applicable not only to the study area of the paper but also to the data sampling frequency. Abrupt changes exceeding those thresholds would be highly unreliable considering that the data sampling period is at most one minute, and usually even less, which are sufficiently high frequencies to not accept more abrupt changes than the ones reflected with those thresholds.

**Minor comments**

**1)**
- **RC1:** *''Introduction: This work also provides air-sea flux data:*
  *Zhang, et al. MASCS 1.0: synchronous atmospheric and oceanic data from a cross-shaped moored array in the northern South China Sea during 2014–2015. Earth System Science Data, 2024, 16(12): 5665-5679.''*

- **AC:** The paper recommended is an interesting example of an observation-based study of air-sea interaction and it has been considered as part of the bibliography references.

- **Changes made:** Line 26: *''studied on offshore platforms, buoys, and vessels in the open ocean since the latest decades of the 20th century (Smith, 1980; Large and Pond, 1981, 1982; Grachev and Fairall, 1997; Edson et al., 2013;.''* changed to *''studied on offshores platforms, buoys, and vessels in the open ocean since the latest decades of the 20th century*

*(Smith, 1980; Large and Pond, 1981, 1982; Grachev and Fairall, 1997; Edson et al., 2013; Zhang et al., 2024)".*

**2)**

- **RC1:** *''Line 14: "process" should be "processes"?''*

- **AC:** Authors agree.

- **Changes made:** Line 15 and 17, *"process"* changed to *''processes''.*

**3)**

- **RC1:** *''Table 5-7: For the "-" in cells without values, the reason should be noted and the format should be uniform.''*

- **AC:** Dashes mean no values available due to lack of measurements. Specified in the caption of Table 5, 6, and 7.

**4)**

- **RC1:** *"Stefan-Boltzman" should be "Stefan-Boltzmann"?*

- **AC:** Authors agree

- **Changes made:** Lines 116, 308 and 369: *''Stefan-Boltzman''* changed to *''Stefan-Boltzmann''.*

**5)**

- **RC1:** *''Page 16: "emmissivity" should be "emissivity"?''*

- **AC:** Authors agree

- **Changes made:** Line 308, *"emmissivity"* changed to *''emissivity''.*

**6)**

- **RC1:** *''Line 200: "makes" should be "make"?''*

- **AC:** *Authors agree*

- **Changes made:** Line 203, *"makes''* changed for *''make''.*

**7)**

- **RC1:** *''All punctuation marks in the text should be uniformly formatted in English.''*

- **AC:** All punctuation marks have been uniformly formatted in English

---

## Author Comment (AC3)

*Additional changes beyond the responses to the referee comments*

- Line 79 : ''Also, different regional wind regimes, so important for the air-sea interaction, are characteristic for each region'' changed to ''Also, different regional wind regimes, crucial for air-sea interaction, are characteristic of each region''.

- *Line 93: ''allowing to measure'' changed to ''allowing them to measure''.*

- *Line 167: ''experimental and technical constrictions'' changed to ''experimental and technical constraints''.*

- *Line 240 : ''November and December months'' changed to ''November and December''.*

- *Line 310: ''The Atlantic Meridional Overturning Circulation (AMOC) accounts for the most of the'' changed to ''The Atlantic Meridional Overturning Circulation (AMOC) accounts for most of the''.*

- *Line 329: ''which contributes to'' changed to ''which contribute to''.*

- Line 342: *''Thus, this heterogeneous character encourages the subdivision in different regions proposed in this study.'' changed to ''Thus, this heterogeneous character encourages the subdivision into different regions proposed in this study.''*

- Line 417: Clayson, C. A., Roberts, J. B., and Bogdanoff, A.: SEAFLUX Version 1: a new satellitebased ocean-atmosphere turbulent flux dataset, Int JClimatol (submitted), 2014. removed, and Curry, J., Bentamy, A., Bourassa, M., Bourras, D., Bradley, E., Brunke, M., Castro, S., Chou, S., Clayson, C., Emery, W., et al.: Seaflux,Bulletin of the American Meteorological Society, 85, 409–424, https://doi.org/10.1175/BAMS-85-3-409, 2004 added instead.
  Also, in line 44, (Clayson et al., 2014) changed to (Curry et al., 2004).